# Design and Simulate Intracranial Support to Guide Maxillo Surgery: A Study Based on Bioengineering

**DOI:** 10.3390/diagnostics13243672

**Published:** 2023-12-14

**Authors:** Maria Giulia Cristofaro, Elvis Kallaverja, Francesco Ferragina, Ida Barca

**Affiliations:** Maxillofacial Surgery Unit, Department of Experimental and Clinical Medicine, “Magna Graecia” University, 88100 Catanzaro, Italy; reikey2003@yahoo.com (E.K.); francesco.ferragina92@gmail.com (F.F.); barca.ida@gmail.com (I.B.)

**Keywords:** navigation system, electromagnetic navigation, image-guided surgery, sinus surgery

## Abstract

Background: Intraoperative navigation allows for the creation of a real-time relationship between the anatomy imagined during diagnosis/planning and the site of surgical interest. This procedure takes place by identifying and registering trustworthy anatomical markers on planning images and using a point locator during the operation. The locator is calibrated in the workspace by placing a Dynamic Reference Frame (DRF) sensor. Objective: This study aims to calculate the localization accuracy of an electromagnetic locator of neuro-maxillofacial surgery, moving the standard sensor position to a different position more suitable for maxillofacial surgery. Materials and Methods: The upper dental arch was chosen as an alternative fixed point for the positioning of the sensor. The prototype of a bite support device was designed and generated via 3D printing. CT images of a skull phantom with 10 anatomical landmarks were acquired. The testing procedure consisted of 10 measurements for each position of the sensor: precisely 10 measurements with the sensor placed on the forehead and 10 measurements with the sensor placed on the bite support device. It also evaluated the localization error by comparing the two procedures. Results: The localization error, when the sensor was placed on the bite support device, was lower in the sphere located on the temporal bone. It was the same in the spheres located on the maxillary bone. The test analysis of the data of the new device showed that it is reliable; the tests are reproducible and can be considered as accurate as the traditional ones. In addition, the sensor mounted on this device has proven to be slightly superior in terms of accuracy and accuracy in areas such as the middle third of the face and jaw. Discussion and Conclusion: The realization of the bite support device allowed the sensor to change position concerning its natural site. This procedure allows us to explore structures, such as the frontal site, which were initially difficult to approach with neuronavigation and improves the approach to midface structures, already studied with neuronavigation. The new calibration, with the position of the sensor on the support device in the same reference points sphere, highlighted the reduction in the location error. We can say that the support proposed in this study lays the foundations for a new navigation approach for patients in maxillofacial surgery, by changing the position of the sensor. It has strong points in improving the localization error for some reference points without determining disadvantages both in the calibration and in the surgical impediment.

## 1. Introduction

Today, surgical navigation is one of the most reliable technologies; it continues to transform surgical interventions into safer and less invasive procedures. In surgery, navigation has stimulated technical progress in both exploratory and interventional procedures in those areas with limited access. The surgeon can know, in real time, where he is moving with the instrument and so can be as accurate as possible. For this reason, the first use of surgical locators is in neurosurgery and maxillofacial surgery, where accuracy must be sub-millimeter [1]. We can classify surgical navigators according to the location technology on which they are based. These systems include sensors and emitters of various types (optical, electromagnetic, etc.) that provide data on the position and orientation of an element in space. The purpose of all localizers is to know, in real time, the position of surgical instruments in the operating space and to make everything visible on a monitor. Pre-operative images will also be present on this monitor [2]. In addition to purely anatomical orientation, intraoperative navigation is also used as a measuring instrument and provides detailed information to surgeons [3]. The position and orientation of surgical instruments are determined by the direct visibility of the surgical field [4]. This study aims to improve the result of the operation by ensuring greater precision and less invasiveness, therefore resulting in fewer post-operative risks for the patient. In the Fiagon neuro navigator (Fiagon GmbH, Hennigsdorf, Germany), the sensor is placed on the patient’s forehead with a sticker, which allows you to calibrate the device. Below is an alternative method of positioning the sensor for maxillofacial surgery, whilst always trying to maintain the accuracy of the navigator. The device developed is an orthodontic bite support with cylindrical support in which we placed the electromagnetic sensor. The bite was designed on a specific phantom, on which we performed all the tests to evaluate its accuracy. To do this, it was necessary to perform a phantom’s CT scan, and we created a dental device using CAD software (AutoCAD LT 2018) (Figure 1).

We created the bite with the use of various software such as 3D Slicer (v4.10.0), Fusion 360 (V2.03174), and FreeCad (V0.17). We also tested if the designed device allowed for accurate localization. In surgery, the Fiagon sensor is positioned (with an adhesive) on the patient’s forehead; this prevents access to anatomical areas such as the frontal sinus. To date, the sensor, fundamental in navigation procedures (its absence would make the procedure non-functional), is positioned at a fixed point in the center of the front. The need to explore areas normally not accessible to navigation (forehead, frontal sinuses, pterygoid fossa, infra-temporal regions, or the lower third of the face) led to the idea of placing the sensor on a different site.

## 2. Materials and Methods

We conducted this prospective study at the Maxillofacial Surgery Unit of the “Magna Graecia” University, Catanzaro, Italy, in cooperation with the biomechatronics department. For neuronavigation, we used the Fiagon electromagnetic locator, very often used in maxillofacial surgery. Through specific software (AutoCAD LT 2018), we have designed a new device on which to place the sensor of the neuronavigator. The model created was 3D-printed and was tested for accuracy during surgical navigation. The device is a dental bite holder, designed on a specific phantom, which has a holder in which we placed the sensor of the electromagnetic navigator [5,6,7]. This created bite must have the characteristic of being stable: the sensor during navigation must be immune to small movements to influence its accuracy.

### 2.1. Design Phase

In this study, we used the open-source 3D Slicer software. It provides tools for the analysis, processing, and three-dimensional visualization of medical images, as well as for the search for image-guided therapies [8]. For the design of the bite support, the CT of the phantom was first imported into 3D Slicer, then we processed the CT in coronal, axial, and sagittal planes. We performed manual segmentation of the upper dental arch volumes using the Model to Label Map. Using the crop volume ROI (region of interest) module, only the desired volume was manually delineated (Figure 2). Finally, we exported the model in the .STL file format.

Through the Fusion 360 software, which allows mesh modeling [9,10], once the STL model had been imported, we carried out the conversion into a solid. To create the bite support device, we used the new .STL file, which was uploaded into the FreeCad software. Using the Work and Design environments it is possible to perform operations on the solid geometry and it is possible to model the object with extrusions; in addition, it is possible to create planar geometry such as rectangular lines b-splines, circular arcs, and other shapes.

### 2.2. D-Model

Due to the complexity of the dental structure of the phantom, we created the bite for the incisors and canines only. We removed the teeth that were not affected by the bite (Figure 3).

The FreeCad Sketcher (V0.17) environment, normally used to create 3D geometries [8,11], was used to model the bite. The result is the bite as shown in Figure 4, resulting from the union of the support with the sensor of the navigator.

The sensor holder is the result of the union of different geometric shapes: two parallelepipeds and a cylinder; we positioned the sensor on the cylinder (Figure 5).

The characteristics are as follows: the first parallelepiped has a length of 7.4 mm, a height of 5 mm, and a width of 35 mm; the second parallelepiped has a length of 7.6 mm, a height of 10 mm, and a width of 5 mm, and it also has an angle of 16°; the cylinder has a radius of 14 mm (radius equal to that of the navigator’s sensor) and a height of 4 mm for greater sensor stability. The object thus created (Figure 6) was exported to an .STL file format for 3D printing and then tested on the phantom to verify the stability properties.

### 2.3. Three-Dimensional Printing

The Fused Deposition Modelling works by taking the model designed using a CAD model, exporting it as an STL file, and uploading it into dedicated software [12,13]. We used the Ultimaker Cura software (v3.3). The CAD model prepared with the software was loaded into the 3D printer: we selected the file to be printed and, once the suitable temperature had been reached (it was set in the software and is based on the type of filament used), the printing began. The 3D printer used for this study is the Ultimaker S5 (Ultimaker, Utrecht, Nederland); the filament used is the TPU95A thread (Ultimaker, reseller—3ditaly Ragusa, Ragusa, Italy).

### 2.4. Device Testing

To assess accuracy during navigation, we performed 10 calibrations with the sensor positioned on the new support and 10 calibrations with the sensor positioned on the front. We measured a series of spheres’ distances for each registration performed and we calculated the distance between a sphere’s position in the CT and the sphere’s position after each calibration. Finally, we compared all the measured distances to verify if the sensor placed on the bite support was accurate enough by comparing them with the distances of the sensor placed on the forehead.

### 2.5. Measurements Made

There were 10 spheres placed on the phantom, each of which had been marked with a number (Figure 7).

For each calibration, we evaluated the error of each sphere, between the position of the pointer and the position of the sphere in the CT, and calculated the measurements of each distance. After calibration, we used the command “Length”; it is present in the Fiagon Navigation Software (Version 3.7) module and allows you to measure the distance in millimeters. We used the spheres placed in different anatomical areas (Figure 8) to verify the accuracy of both sensors, the one placed on the front and the one placed on the created support.

### 2.6. Fiagon Measures

For each calibration performed, we calculated seven distances and the location error of each sphere. We calculated the length in millimeters between each sphere with the integrated software of the neuronavigator (Fiagon Navigation Software Version 3.7). The electromagnetic navigator allowed us to take screenshots of each measurement performed (Figure 9 and Figure 10).

In addition to the distances between the spheres, we have calculated the distance between the spheres in the CT and the real-time position of the pointer on the surface [14]. For all 10 spheres and 20 calibrations performed, we also calculated the location error resulting from the navigator.

### 2.7. Sphere Localization Error

For each calibration performed and for all 10 spheres on the phantom’s surface, we measured the error after calibration between the position of the i-th sphere visible in the CT and the real-time position of the navigator pointer placed on the spheres’ surface. When the location error was not significant, we set a value of 0 mm. This error is called the Target Registration Error (TRE) (Figure 11) and it determines the distance between the corresponding spheres after registration.

The TRE is the relevant measure used to estimate the accuracy of the navigator. It defines the distance between the reference point on the TC data set and the coordinates specified by the electromagnetic tracking system when the pointer is on the reference sphere [15,16]. By processing the calculated measurements, we estimated which sphere had a greater error than all the recordings made. In addition, the error depends on the anatomical area where the spheres were placed, and which sensor was used.

By calculating the distances between the spheres, it is possible to estimate the accuracy of the navigator with the two different sensors used. For data processing, we used the Python programming language. We took the median and interquartile as references to study the results obtained. The interquartile is a dispersion index, which is a measure of how much the values differ from a central value. We used special plots, called box plots, to display both the median and the interquartile. The box plot has become the standard technique for presenting the summary of five important parameters for each distribution, which includes the minimum and maximum interval value, the upper and lower quartiles, and the median. This collection of values is a quick way to summarize the distribution of a set of data and this reduced representation is the easiest way to compare them [17]. Box plots also display anomalous values: values that are at an abnormal distance from others.

Using the Jupyter application (v5.3.4), which uses Python (v3.6.8) as its programming language, we created marked vectors related to the 10 spheres taken as objects. Each sphere is a vector composed of 10 errors calculated for all 10 calibrations. We used the box plot command to display the median and interquartile values, inserting the variables related to errors. Each sphere will have a different error based on the calibration performed. We also performed an analysis for the calibrations. Then we created vectors for calibration; each vector reported all the errors of the 10 spheres, inherent to the relative calibration.

### 2.8. Statistical Analysis

We performed descriptive statistical analyses on the recorded data, using the categorical data’s central tendency indices and absolute and relative frequencies. An operator performed all the calibrations; hence, we calculated the intra-rater reliability test, precisely, the consistency of the results using the Pearson correlation coefficient. We accepted a statistical significance at *p* < 0.05. We performed the analysis using GraphPad Prism (v8.0.0), statistical software (GraphPad Company, San Diego, CA, USA).

## 3. Results

We present the results of the localization error of each sphere for each calibration performed (Table 1 and Table 2).

### 3.1. Sphere Localization Error for the Sensor Placed on the Forehead

Table 1 shows the Position Errors of each sphere for the 10 calibrations performed with the sensor positioned on the forehead.

We compared the median and interquartile values, as in Figure 12. The box plots show an error of 0 mm, regarding spheres number 1, 3, 5, and 8. This means that they are found almost optimally in almost all the calibrations performed. The calculated median and interquartile are both 0 mm. The spheres with the greatest localization error are the numbers 4 and 5 (the spheres placed on the temporal bone of the phantom), with a median of 4 mm and 3.93 mm, respectively. Spheres number 9 and 10 also show significant errors with a median value of 2.6 mm and 2.97 mm. The interquartile ranges are greater for spheres number 9 and 10, presenting greater variability than the others. From the results of the median and the interquartile for each one (Figure 13), no variability is noted for calibrations number 3 and 4, while if the interquartile is considered, there are quite variable values up to 1.66 mm.

### 3.2. Sphere Localization Error for the Sensor Placed on the Bite

Table 2 shows the errors of each sphere when calibrations are performed with the sensor positioned on the bite.

The box plots of Figure 14 and Figure 15 show spheres number 1, 3, and 4 without errors in all the calibrations with median and interquartile equal to 0 mm. Sphere number 10, on the other hand, has a high localization error while the median is 3.06 mm. In calibrations number 9 and 10, the median values are higher than the others. However, calibrations number 5 and 6 have upper interquartile ranges of 1.73 mm and 1.57 mm.

We confirmed significance with a value of *p* = 0.0025 with a value r = 0.2996; the 95% confidence interval is between 0.1096 and 0.4684.

### 3.3. Fiagon Distance Measurements

For both sensors (on the forehead and the bite), we calculated the seven distances between the various spheres (indicated with the letters a–g). We made ten calibrations for each of these. Table 3 shows all lengths calculated in millimeters for the sensor on the forehead.

Similarly, we performed the 10 calibrations with the sensor on the stand; all measured distances are shown in Table 4.

The calculated distances were compared to understand how much the accuracy of the navigator is maintained with the sensor positioned on the support. Calculating the distances between the lead spheres helps to further study the navigational accuracy when the sensor is not on the forehead. Therefore, comparing distances in both configurations is another method to estimate the accuracy of navigation.

We confirmed significance with a value of *p* < 0.0001 with a value r = 0.9991; the 95% confidence interval is between 0.9986 and 0.9995.

### 3.4. Three-Dimensional Slicer Distance Measurements

The 3D Slicer software was used to take a value of the distances between the spheres in the CT, not affected by the inherent calibration error. The spheres have a diameter of 2.5 mm, while the thickness of the TC slice is 0.5 mm, so each sphere will be on more than one slice and for this reason, the spheres have been centered. After calculating the seven distances examined, they were repeated five times for reliable distance measurement. We made an average between each repeated distance. Table 5 presents all calculated values and the relative average of all seven distances. This calculation is essential to obtain a real reference of the distances between the spheres. Given this value, we calculated the difference for each distance with the relative distances calculated after calibration for both sensors, to obtain an estimate of the error and evaluate its accuracy.

### 3.5. Distances between the Spheres

The distance measurements calculated using Slicer were useful for estimating the error of the distances between the spheres. For all the distances and related calibrations, we used the difference between the average distance calculated using Slicer and the i-th distance measured by the navigator (Equation (1)).
Estimated Error = D (s) − D (n)(1)
where D (s) is the average of the i-th distance calculated using Slicer; D (n) is the i-th distance calculated by the navigator.

Table 6 and Table 7 report the errors relating to the distances taken as a reference to the calculation estimated on Slicer. An error was estimated for all the distances calculated by the Fiagon, for both sensors.

We confirmed significance with a value of *p* < 0.0001 and with a value r = 0.4790; the 95% confidence interval was between 0.2750 and 0.6418.

### 3.6. The Error between the Distance Measurement Calculated on the Slicer and the Distance Measurement of the Sensor Placed on the Forehead

Here, reference is made to Table 6, which describes the errors calculated by taking as reference the averages of the distances calculated on the Slicer. From the box plots of Figure 16, we evaluated the results of the median and the distance F has the smallest error compared with the others, equal to 0.042 mm, while the greatest error concerns distance C with a median equal to −2.057 mm. The interquartile calculation did not return significantly conflicting values. Each distance was a vector since each distance will have the 10 values calculated for all 10 calibrations. We also calculated the calibration error, when the sensor was present on the forehead connected to the Fiagon. Figure 17 shows the relative box plot. We deduced that the first calibration had a lower error when compared with the others, while the one with a greater error was 9. The highest interquartile values concern calibrations 5, 6, and 7, with a greater variability concerning the others.

### 3.7. The Error between the Distance Measurement Calculated on the Slicer and the Distance Measurement of the Sensor Placed on the Bite Support

In this case, we refer to the errors calculated in Table 7; we created the vectors relating to the seven distances considering the ten calibrations carried out with the sensor placed on the support. Based on the box plot of Figure 18, we affirm that distance E is the one with the lowest errors, with a median calculated as −0.065 mm. The greatest error is present in distance C, with a median equal to −1.87 mm. As regards the relative interquartile range, there is no significant variability. As for the calibration error (Figure 19), the median values indicate that the error is almost constant for all 10 calibrations, and there are no “abnormal” values. The higher interquartile values are for calibration 4 and calibration 10.

### 3.8. The Error between the Distance of the Two Sensors

To understand how similar the distances calculated by the two sensors were after the calibration, we carried out an analysis. These considerations are useful for understanding whether the support created is sufficiently precise. For this operation, we calculated the difference again between the i-th distances of Table 3 and the i-th distances of Table 4. The results obtained are summarized in Table 8.

The box plot was used to visualize the median and interquartile values, inserting the variables relating to the errors, to be able to compare them in Figure 20. By comparing the median values, it can be stated that the distance with the greatest error is the E, with a median equal to 0.135 mm; it is also the distance that presents a greater variability of error with the same interquartile at 0.91 mm. The errors relating to distance A and distance B are almost unchanged in the case of the two sensors, with a median equal to −0.035 mm and 0.005 mm, respectively. In principle, there are no significant differences between the distances calculated with the sensor placed on the forehead and with the sensor placed on the bite support. In the same way, we evaluated the error by comparing all the calibrations, to evaluate if there is any calibration that differs particularly from the others. Once we created the corresponding vectors, relating to the rows of Table 8, we displayed the box plots relating to all the calibrations in Figure 21, and we recalculated the median and the interquartile of each. There is no significant variability between the 10 calibrations. The highest interquartile ranges were for calibration 4 and calibration 5, which had a greater variability than the others. The median had higher values for calibration 10. We analyzed the data for calibration since there is subjective variability; in each calibration, there was an error made by the operator.

### 3.9. Final Remarks

By carrying out the first comparison, it is possible to estimate which spheres are found to result in the least error, in both sensors. The number 4 and 5 spheres, located close to the temporal bone of the phantom, are located without errors when the sensor is placed on the bite support. Conversely, they result in a major error when the sensor is placed on the forehead. Spheres 1 and 3 are well positioned in both cases. Spheres 9 and 10 show relevant errors in both cases, this suggests that with the two sensors, some anatomical parts of the phantom are located better than others. We evaluated the error of the calculated distances between the spheres for the two electromagnetic sensors and whether the values differ with a non-parametric Wilcoxon test.

### 3.10. Wilcoxon Test

The Wilcoxon test is one of the most important non-parametric tests for verifying, in the presence of ordinal values from a continuous distribution, whether two statistical databases come from the same population. The Wilcoxon test is for non-independent databases and, unlike means and medians, its value will have a one-to-one correspondence with the result of the Wilcoxon rank sum test [18,19,20]. Using the “Scipy” Python library, it was possible to perform the non-parametric test of Wilcoxon which, based on a calculated *p*-value and given two distributions, specifies whether they have the same or different distributions. Considering the results of the medians corresponding to the distance error of the forehead sensor and the bite support, the result is that the distribution was the same. Therefore, the values of the distributions can be comparable and, from this, it can be concluded that the support created specifically for the sensor is quite accurate also considering the box plots of Figure 22 and Figure 23. There are no significant differences in the distances calculated with the sensor placed on the forehead. Concerning the sphere localization error in the case of the two sensors, the calculated median results were examined, and a Wilcoxon test was performed. The result of the latter led to a different distribution. This result is because some spheres are located without errors by the sensor placed on the forehead and others with errors. From the considerations expressed up to now, it is possible to state that the sensor on the designed support is quite accurate when compared with the results of the sensor placed on the forehead. To confirm these results, we also demonstrated how some spheres, such as 4 and 5, are found without errors when placing the sensor on the bite support.

## 4. Discussion

In the environment of surgical navigation, there is a need to locate surgical instruments and anatomical structures in real time, to ensure less invasiveness and higher precision by the surgeon. It is interesting how, in recent years, artificial intelligence has been growing in the diagnosis of craniofacial pathologies; a starting point could also be the integration of this technology in combination with neuronavigation [21,22,23]. In the future, it will be interesting to understand how much this intelligence will help us to be precise and meticulous in surgical procedures. The Fiagon electromagnetic navigator involves the use of a sensor, positioned on the patient’s forehead. This study aims to create a device with a cylindrical support on which to place the sensor in question. Once the stability of the bite with its support was verified, it was tested on a phantom, testing its accuracy and precision. To do this, the phantom was placed at different anatomical points, as were a series of ten lead spheres of diameter φ = 2.5 mm, which did not interfere with the navigator’s magnetic field generator. Subsequently, we performed both the CT of the phantom with the spheres and a series of calibrations (10 for the sensor positioned on the forehead and 10 for the sensor positioned on the bite support) [24,25,26]. Measuring the corresponding distances between the spheres with the navigator pointer after each calibration, a series of measurements were also made that describe the localization errors of each sphere. After a statistical analysis of the results obtained, it is possible to state that the navigator used with the sensor placed on the device is quite accurate when compared with the sensor placed on the forehead. The results found show that the measurements of the distances between the spheres do not have great variability between the two configurations, while with regards to the localization error of the spheres, some of these are optimally located from one sensor to the other. This suggests the importance of the position of the sensor in locating the different anatomical areas. However, it must be remembered that CT errors and registration errors play a key role in the accuracy of the device. The spheres with the best location in both combinations were spheres 1 and 3 while the sphere located with the worst error was 10, placed on the bite support. The most relevant information concerns spheres 4 and 5, which resided in the temporal bone and were located almost without errors with the sensor placed on the support bite and also on the forehead. Therefore, the position of the sensor influences the localization and, according to various hypotheses, it may have depended on the magnetic field generator that is located below the phantom. However, these tests demonstrate the importance of the device created, since the anatomical areas are localized without errors when the sensor is placed on the support [27]. Using this device, even if the sensor is not directly on the phantom, it is possible to conduct navigation without relevant errors. Later, other tests can be carried out to support the hypothesis written above on how much the position of the sensor affects the localization of anatomical points. Concerning maxillofacial surgery, it is possible to state that the designed device is a good alternative to placing the sensor and is more accurate when you want to locate the patient’s temporal bone, frontal sinus, and pterygoid fossa. With the use of more spheres, it will also be possible to determine with greater accuracy which are exactly the points where you are navigating without significant errors compared with the sensor on the forehead.

## 5. Conclusions

The studied device is both an engineering and a medical invention. Today, there are still few uses in surgery, especially in maxillofacial surgery. The proposed modifications promote several ideas for future studies. Neuronavigation is a fairly recent technology that allows us to plan the preoperatory phase and set up an image-guided intervention strategy. The proposed device allows for the allocation of the tracking sensor in an anatomical district (oral cavity) different from the front. This new position allows the exploration of anatomical regions (both soft tissues and hard tissues; upper face, middle face, temporal fossa, maxilla, and oral cavity) that are difficult to access today. By analyzing the TRE (to estimate the accuracy of the navigator) and calculating the measurements, it emerged that between the two examined configurations (sensor placed on the forehead and the bite), some spheres are better than others and some spheres are worse. After the various analyses, we sustain that the sensor placed on the device, compared with the one placed on the front, is significantly more accurate in some places and is equally accurate in other places. This allows the localization of anatomical areas with greater precision and exceeds the current limits of tracking devices. After carrying out the tests, we can state that the designed device has advantages for maxillofacial surgery.

## Figures and Tables

**Figure 1 diagnostics-13-03672-f001:**
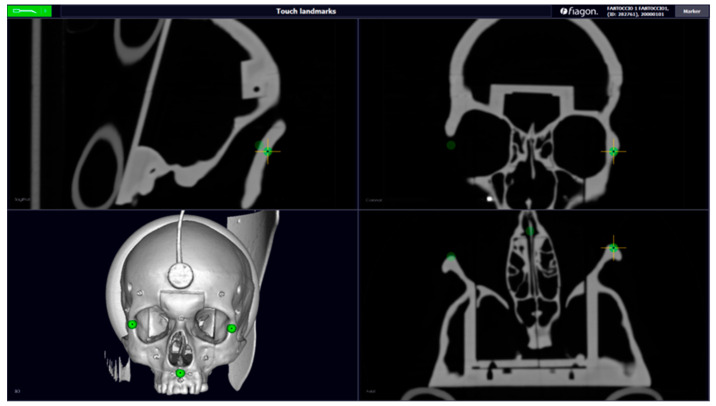
CT scanning (axial, sagittal, coronal, and 3D) with the dental device, created using CAD software. Green points—Fiducal Landmarks for calibration (maxillo malar suture and anterior nasal spine).

**Figure 2 diagnostics-13-03672-f002:**
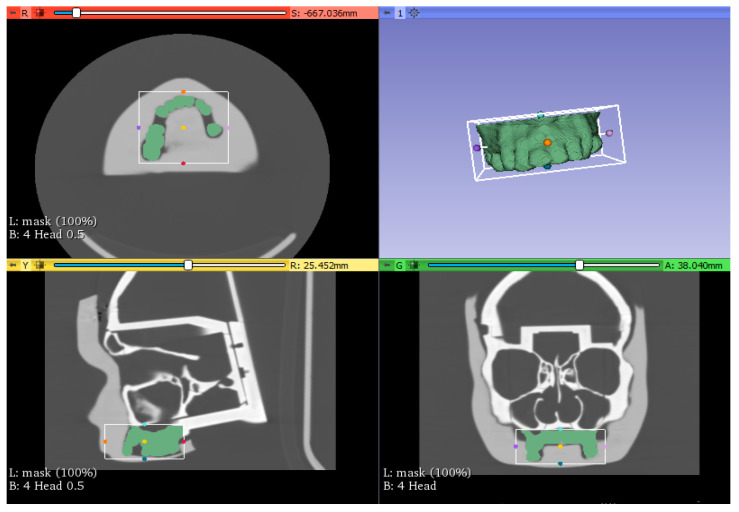
Volume Cropping.

**Figure 3 diagnostics-13-03672-f003:**
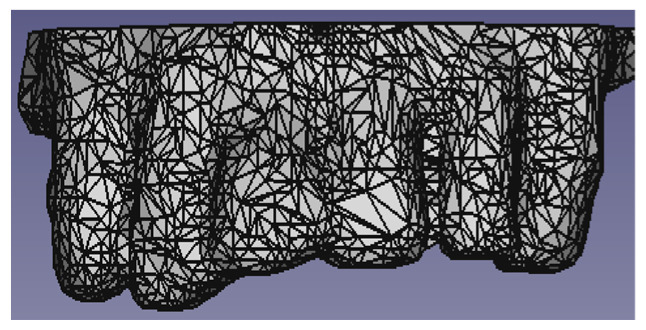
Dental structure of the phantom.

**Figure 4 diagnostics-13-03672-f004:**
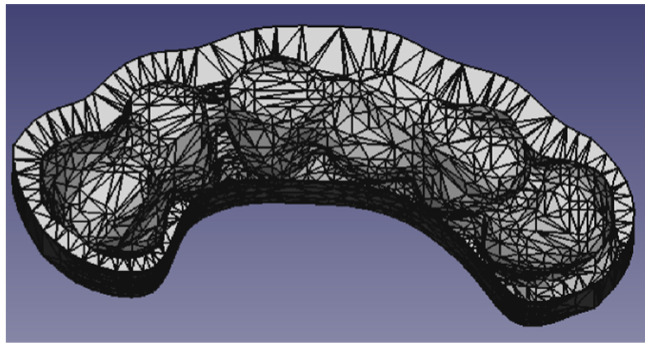
Bite obtained from the phantom’s upper dental structure.

**Figure 5 diagnostics-13-03672-f005:**
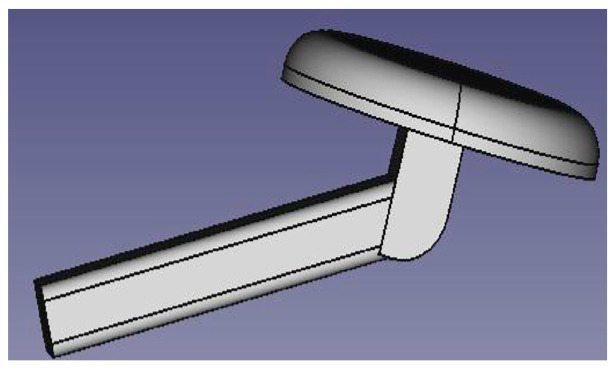
Sensor support.

**Figure 6 diagnostics-13-03672-f006:**
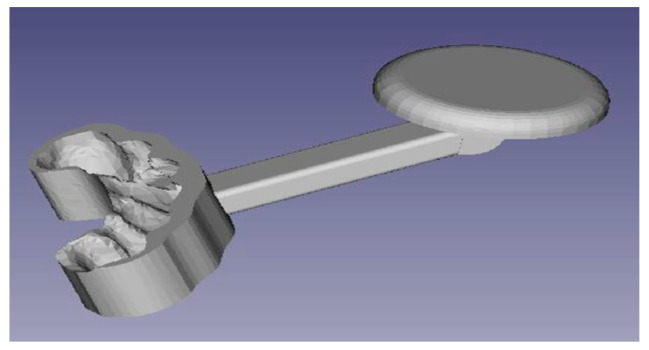
Prototype.

**Figure 7 diagnostics-13-03672-f007:**
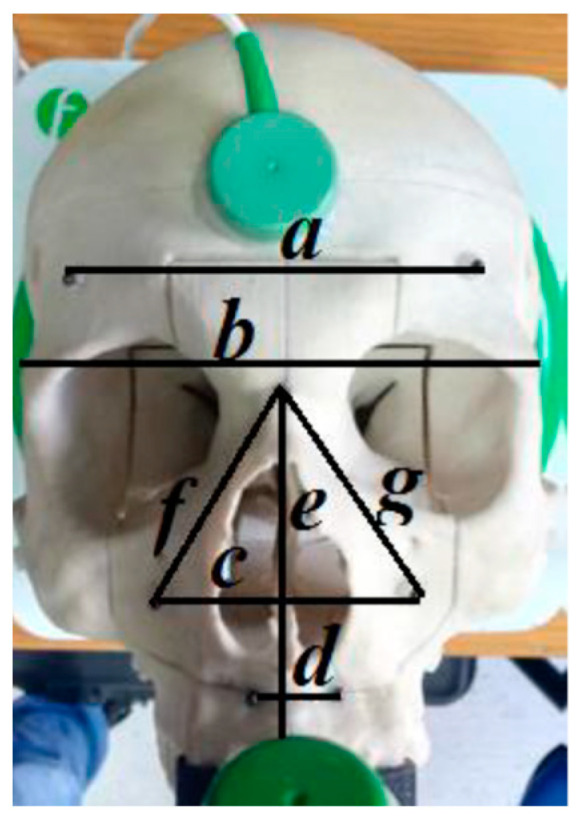
Distance between the spheres is indicated by letters. The distances between the spheres measured with the software: (a) frontal right–left, (b) temporal right–left, (c) maxillary right–left, (d) premaxilla right–left, (e) glabella–anterior nasal spine, (f) glabella–right maxilla, (g) glabella–left maxilla.

**Figure 8 diagnostics-13-03672-f008:**
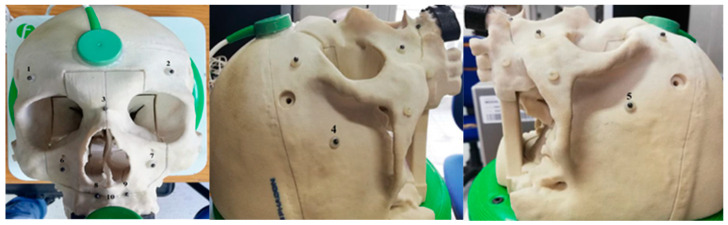
Spheres marked with numbers.

**Figure 9 diagnostics-13-03672-f009:**
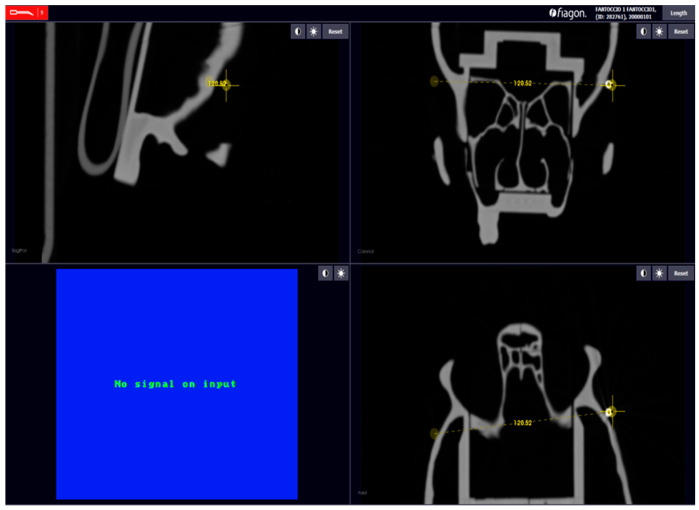
Example: measure distance.

**Figure 10 diagnostics-13-03672-f010:**
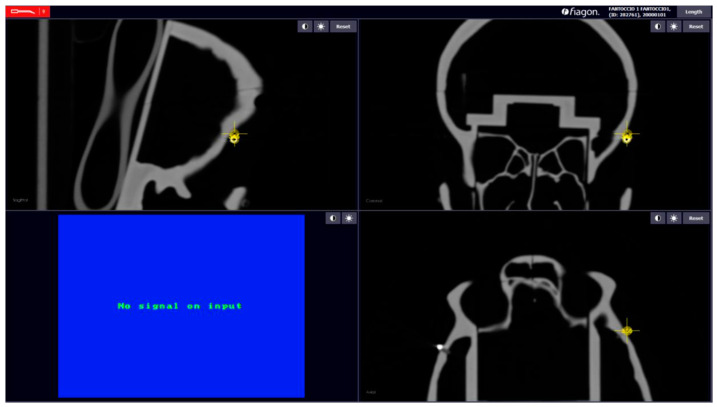
Error localization sphere nr.5.

**Figure 11 diagnostics-13-03672-f011:**
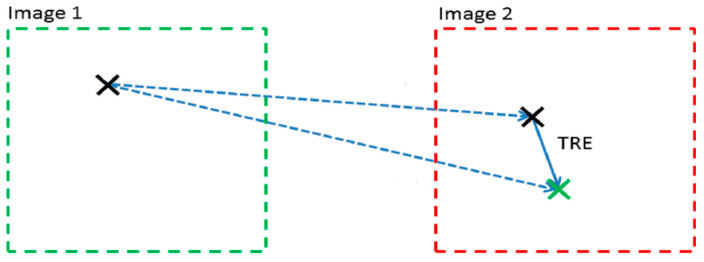
Target Registration Error. Black ‘X’ is the sphere shown on the TC scan. Green ‘X’ is the real-time position of the navigator pointer located on the surface of the sphere.

**Figure 12 diagnostics-13-03672-f012:**
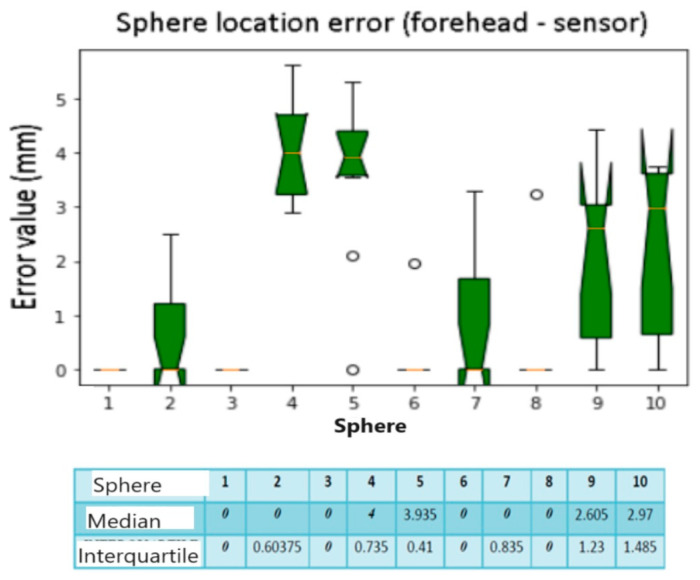
Box plot sphere location error. The bars indicate the minimum and maximum values of a certain range, without outliers. The orange line indicates the median of the data and gives an idea of the central error trend. Instead, the white bullets indicate the outliers.

**Figure 13 diagnostics-13-03672-f013:**
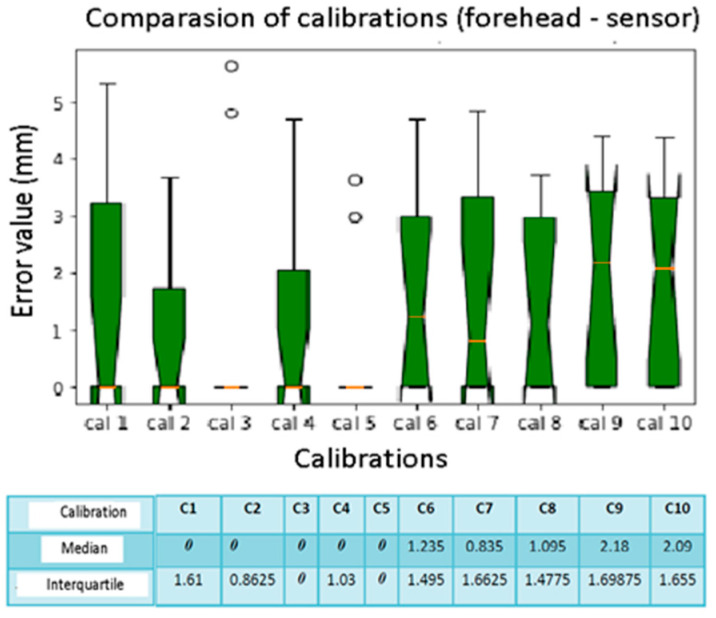
Box plot comparison of calibration. The bars indicate the minimum and maximum values of a certain range, without outliers. The orange line indicates the median of the data and gives an idea of the central error trend. Instead, the white bullets indicate the outliers.

**Figure 14 diagnostics-13-03672-f014:**
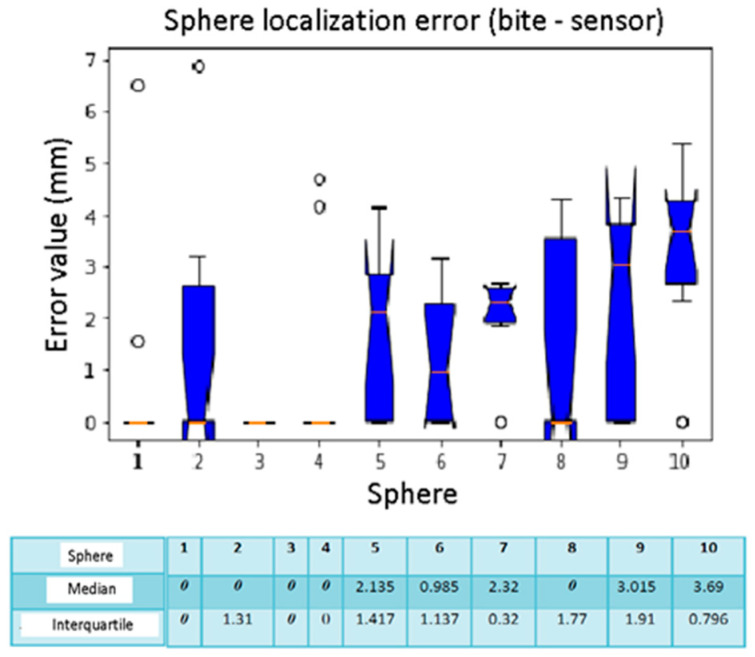
Box plot sphere location error. The bars indicate the minimum and maximum values of a certain range, without outliers. The orange line indicates the median of the data and gives an idea of the central error trend. Instead, the white bullets indicate the outliers.

**Figure 15 diagnostics-13-03672-f015:**
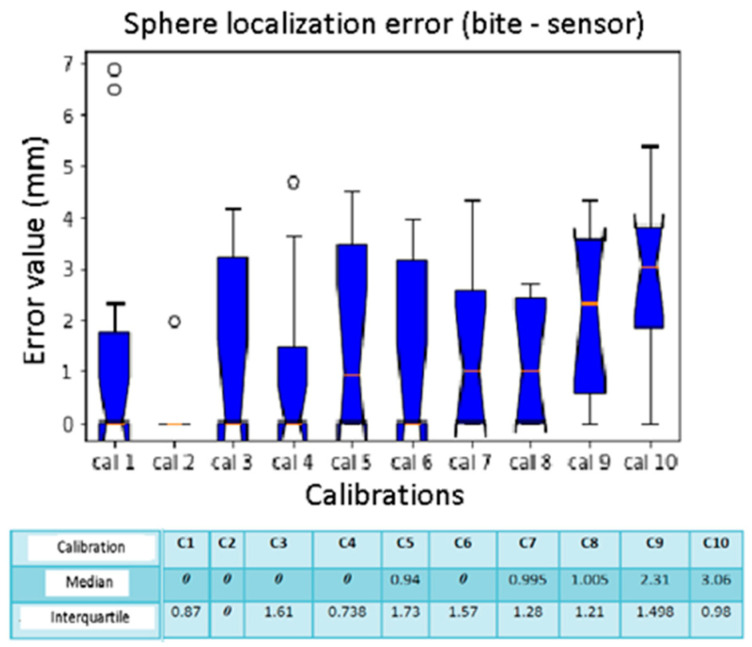
Box plot comparison of calibration. The bars indicate the minimum and maximum values of a certain range, without outliers. The orange line indicates the median of the data and gives an idea of the central error trend. Instead, the white bullets indicate the outliers.

**Figure 16 diagnostics-13-03672-f016:**
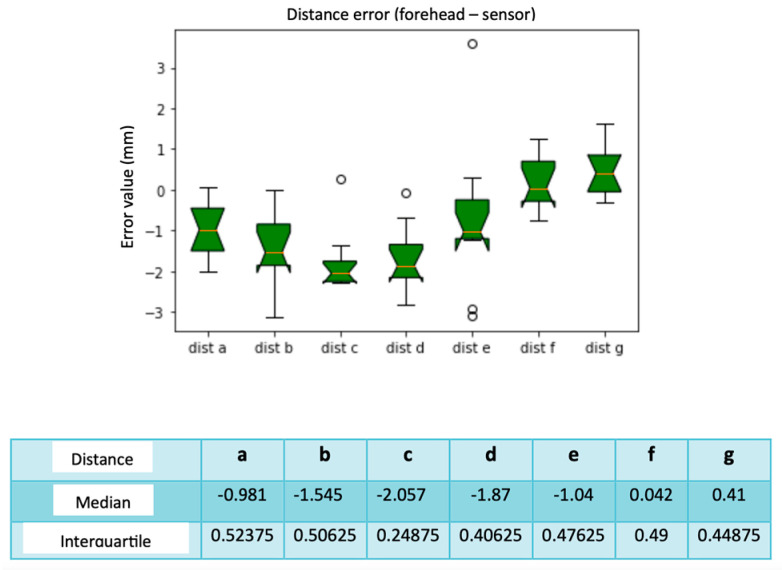
Box plot distance error. The bars indicate the minimum and maximum values of a certain range, without outliers. The orange line indicates the median of the data and gives an idea of the central error trend. Instead, the white bullets indicate the outliers.

**Figure 17 diagnostics-13-03672-f017:**
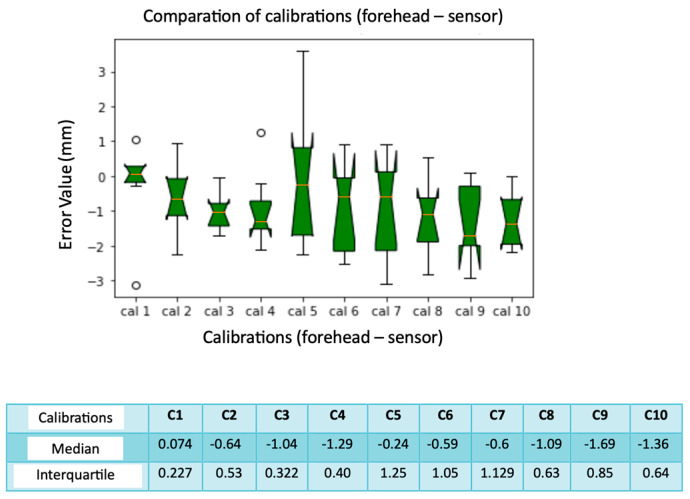
Box plot comparison of calibration. The bars indicate the minimum and maximum values of a certain range, without outliers. The orange line indicates the median of the data and gives an idea of the central error trend. Instead, the white bullets indicate the outliers.

**Figure 18 diagnostics-13-03672-f018:**
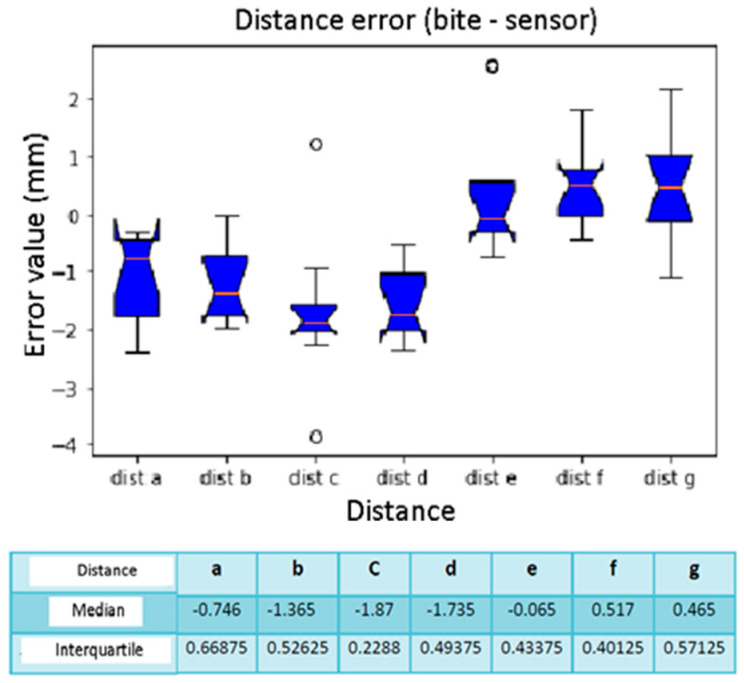
Box plot distance error. The bars indicate the minimum and maximum values of a certain range, without outliers. The orange line indicates the median of the data and gives an idea of the central error trend. Instead, the white bullets indicate the outliers.

**Figure 19 diagnostics-13-03672-f019:**
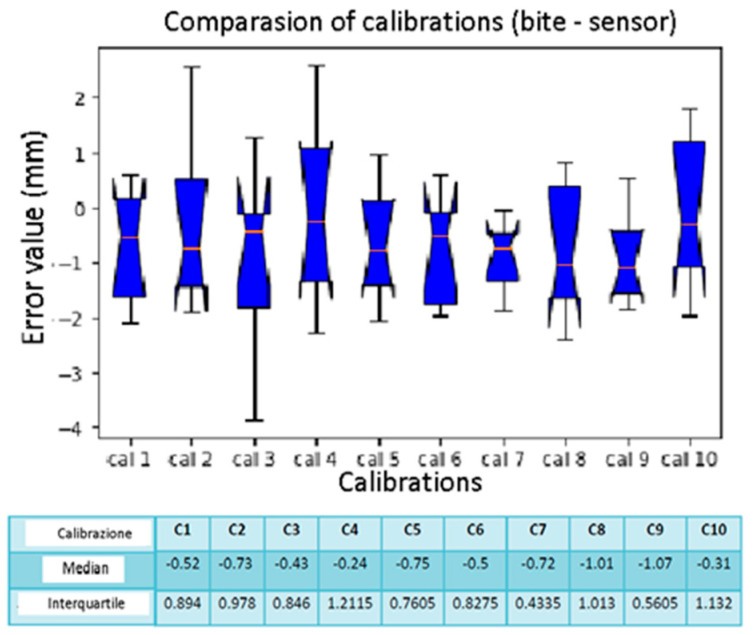
Box plot comparison of calibration. The bars indicate the minimum and maximum values of a certain range, without outliers. The orange line indicates the median of the data and gives an idea of the central error trend. Instead, the white bullets indicate the outliers.

**Figure 20 diagnostics-13-03672-f020:**
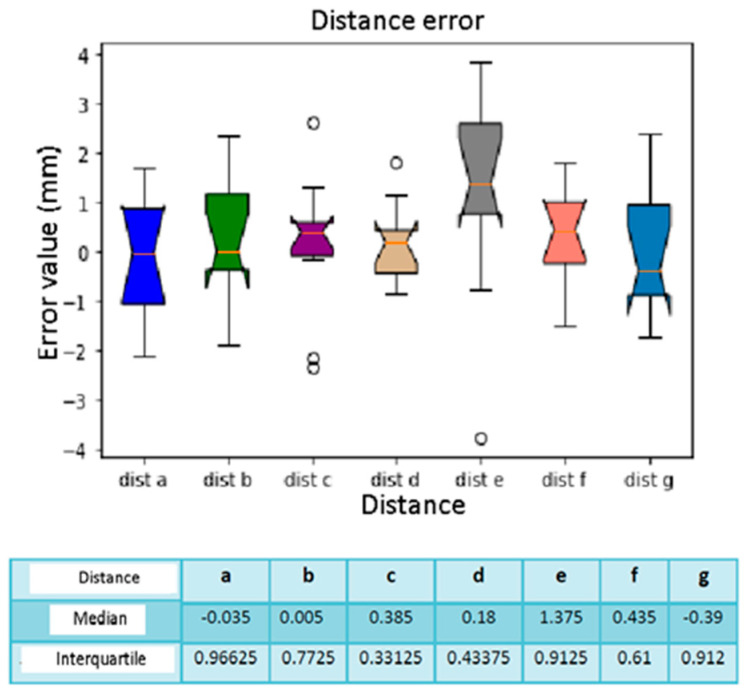
Box plot distance error. The bars indicate the minimum and maximum values of a certain range, without outliers. The orange line indicates the median of the data and gives an idea of the central error trend. Instead, the white bullets indicate the outliers.

**Figure 21 diagnostics-13-03672-f021:**
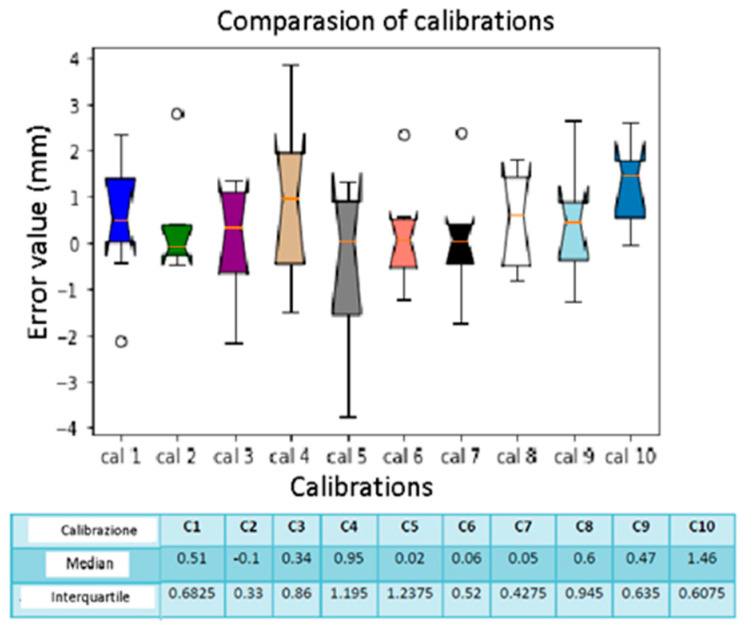
Box plot comparison of calibration. The bars indicate the minimum and maximum values of a certain range, without outliers. The orange line indicates the median of the data and gives an idea of the central error trend. Instead, the white bullets indicate the outliers.

**Figure 22 diagnostics-13-03672-f022:**
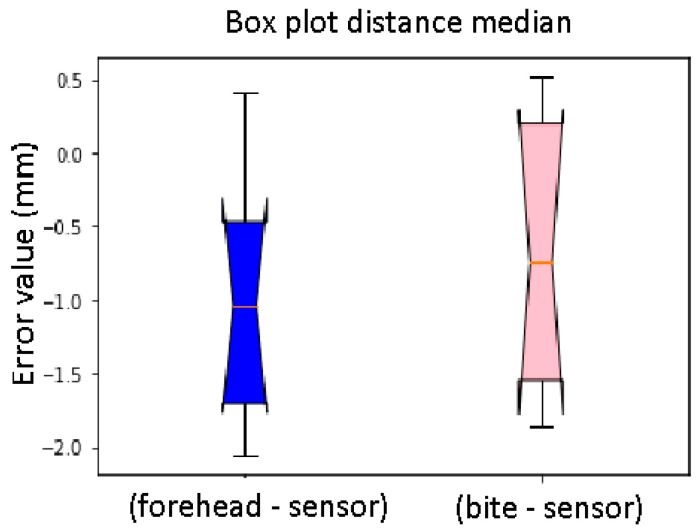
Box plot distance median. Difference between the sensor on the forehead and the bite support (distance). The bars indicate the minimum and maximum values of a certain range, without outliers. The orange line indicates the median of the data and gives an idea of the central error trend.

**Figure 23 diagnostics-13-03672-f023:**
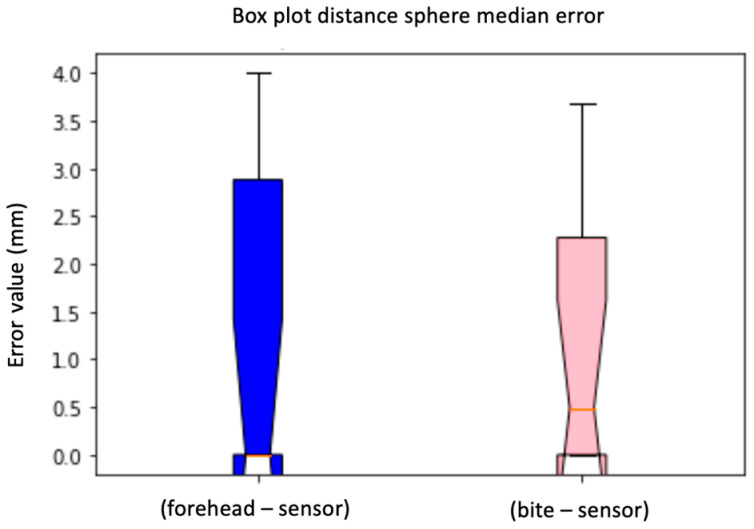
Box plot distance sphere median error. Difference between the sensor on the forehead and the bite support (sphere). The bars indicate the minimum and maximum values of a certain range, without outliers. The orange line indicates the median of the data and gives an idea of the central error trend.

**Table 1 diagnostics-13-03672-t001:** Position Error when the sensor is placed on the forehead.

Reference Sphere	1	2	3	4	5	6	7	8	9	10
Calibration 1	0	0	0	3.16	5.3	0	0	3.23	4.42	0
Calibration 2	0	0	0	3.1	3.67	0	0	0	2.3	0
Calibration 3	0	0	0	5.62	4.79	0	0	0	0	0
Calibration 4	0	0	0	4.69	2.09	1.97	0	0	0	3.63
Calibration 5	0	0	0	3.63	0	0	0	0	0	2.97
Calibration 6	0	0	0	4.7	4.14	0	2.47	0	3.13	2.57
Calibration 7	0	0	0	4.84	3.55	0	1.67	0	2.65	3.76
Calibration 8	0	2.19	0	2.91	3.73	0	0	0	3.26	2.97
Calibration 9	0	1.61	0	3.43	4.41	0	3.3	0	2.75	3.75
Calibration 10	0	2.51	0	4.37	4.37	0	1.67	0	2.56	3.56

**Table 2 diagnostics-13-03672-t002:** Position Error when the sensor is placed on the bite.

Reference Sphere	1	2	3	4	5	6	7	8	9	10
Calibration 1	6.51	6.87	0	0	0	0	2.32	0	0	0
Calibration 2	0	0	0	0	1.97	0	0	0	0	0
Calibration 3	0	0	0	4.17	4.12	0	2.35	0	0	3.5
Calibration 4	0	0	0	4.69	0	1.97	0	0	0	3.63
Calibration 5	0	0	0	0	0	1.99	1.88	4.21	3.95	4.51
Calibration 6	0	0	0	0	3.3	0	2.65	0	3.45	3.96
Calibration 7	0	0	0	0	0	2.37	2.63	1.99	3.43	4.34
Calibration 8	0	2.72	0	0	2.43	0	2.01	0	2.6	2.37
Calibration 9	0	2.29	0	0	2.3	3.03	2.32	4.32	4.34	3.75
Calibration 10	1.56	3.2	0	0	2.97	3.16	2.57	4.06	4.01	5.39

**Table 3 diagnostics-13-03672-t003:** Distance measurements when the sensor is placed on the forehead.

Distance	a	b	c	d	e	f	g
Calibration 1	83.19	121.89	50.53	16.06	62.5	50.51	49.96
Calibration 2	83.9	120.36	53.06	16.65	62.46	51.47	49.29
Calibration 3	84.61	119.57	52.5	17.5	63.26	52.32	50.29
Calibration 4	84.82	120.25	52.9	17.27	63.46	50.29	50.44
Calibration 5	84.59	118.78	53.04	18.05	58.63	51.79	48.6
Calibration 6	83.67	121.27	52.73	18.36	61.93	52.14	49.33
Calibration 7	83.69	119.36	53.07	17.99	65.33	50.65	49.58
Calibration 8	85.04	119.73	52.81	18.8	63.31	51.85	49.7
Calibration 9	83.81	120.45	53.07	17.71	65.15	51.47	50.56
Calibration 10	85.26	120.68	52.16	18.17	63.26	51.55	50.56

**Table 4 diagnostics-13-03672-t004:** Distance measurements when the sensor is placed on the bite.

Distance	a	b	c	d	e	f	g
Calibration 1	85.32	119.97	52.88	16.5	61.68	50.94	50.47
Calibration 2	84	120.52	52.7	17.1	59.65	51.07	49.69
Calibration 3	83.57	118.78	54.65	18.35	62.17	51.98	48.95
Calibration 4	84.59	120.25	53.01	17.27	63.16	50.29	50.11
Calibration 5	84.02	120.68	51.72	18.03	62.4	50.57	49.86
Calibration 6	84.9	118.94	52.67	17.93	62.72	51.57	49.62
Calibration 7	83.66	119.31	52.66	17.6	62.94	51.59	51.3
Calibration 8	85.66	120.53	52.35	16.99	61.65	50.73	50.08
Calibration 9	85.1	120.22	52.46	16.57	62.51	51	51.31
Calibration 10	83.57	120.72	49.57	17.83	62.53	49.74	49.1

**Table 5 diagnostics-13-03672-t005:** Distance measurements with 3D Slicer software.

Distance	a	b	c	d	e	f	g
Measure 1	82.82	118.7	51	15.9	61.8	52.4	50.06
Measure 2	83.7	118.8	51.4	16.3	61.4	52.5	50.02
Measure 3	82.9	119.2	51	15.9	63	51.4	51
Measure 4	83.3	118.4	50.5	16.3	62.3	50.06	50.05
Measure 5	83.6	118.7	50.09	15.5	62.6	51.4	50.7
Average	83.264	118.76	50.798	15.98	62.22	51.552	50.24

**Table 6 diagnostics-13-03672-t006:** Registration error: sensor on the forehead.

Distance	a	b	c	d	e	f	g
Calibration 1	0.074	−3.13	0.268	−0.08	−0.28	1.042	0.28
Calibration 2	−0.636	−1.6	−2.262	−0.67	−0.24	0.082	0.95
Calibration 3	−1.346	−0.81	−1.702	−1.52	−1.04	−0.768	−0.05
Calibration 4	−1.556	−1.49	−2.102	−1.29	−1.24	1.262	−0.2
Calibration 5	−1.326	−0.02	−2.242	−2.07	3.59	−0.238	1.64
Calibration 6	−0.406	−2.51	−1.932	−2.38	0.29	−0.588	0.91
Calibration 7	−0.426	−0.6	−2.272	−2.01	−3.11	0.902	0.66
Calibration 8	−1.766	−0.97	−2.012	−2.82	−1.09	−0.298	0.54
Calibration 9	−0.546	−1.69	−2.272	−1.73	−2.93	0.082	−0.32
Calibration 10	−1.996	−1.92	−1.362	−2.19	−1.04	0.002	−0.32

**Table 7 diagnostics-13-03672-t007:** Registration error: sensor on the bite support.

Distance	a	b	c	d	e	f	g
Calibration 1	−2.056	−1.21	−2.082	−0.52	0.54	0.612	−0.23
Calibration 2	−0.736	−1.76	−1.902	−1.12	2.57	0.482	0.55
Calibration 3	−0.306	−0.02	−3.852	−2.37	0.05	−0.428	1.29
Calibration 4	−1.326	−1.49	−2.242	−1.29	−1.24	1.262	−0.2
Calibration 5	−0.756	−1.92	−0.922	−2.05	−0.18	0.982	0.38
Calibration 6	−1.636	−0.18	−1.872	−1.95	−0.5	−0.018	0.62
Calibration 7	−0.396	−0.55	−1.862	−1.62	−0.72	−0.038	−1.06
Calibration 8	−2.396	−1.77	−1.552	−1.01	0.57	0.822	0.16
Calibration 9	−1.836	−1.46	−1.662	−0.59	−0.29	0.552	−1.07
Calibration 10	−0.306	−1.96	1.228	−1.85	−0.31	1.812	1.14

**Table 8 diagnostics-13-03672-t008:** Distance error between the sensor on the forehead and the bite support.

Distance	a	b	c	d	e	f	g
Calibration 1	−2.3	1.92	−2.35	−0.44	0.82	−0.43	−0.51
Calibration 2	−0.1	−0.16	0.36	−0.45	2.81	0.4	−0.4
Calibration 3	1.04	−0.46	−2.15	−0.85	1.09	0.34	1.34
Calibration 4	0.95	1.47	−0.14	−0.79	3.83	−1.5	2.38
Calibration 5	0.57	−1.9	1.32	0.02	−3.77	1.22	−1.26
Calibration 6	−1.23	2.33	0.06	0.43	−0.79	0.57	−0.29
Calibration 7	0.03	0.05	0.41	0.39	2.39	−0.94	−1.72
Calibration 8	−0.62	−0.8	0.46	1.81	1.66	1.12	−0.38
Calibration 9	−1.29	0.23	0.61	1.14	2.64	0.47	−1.02
Calibration 10	1.69	−0.04	2.59	0.34	0.73	1.81	1.46

## Data Availability

The datasets generated and analyzed during the study are available from the corresponding author on reasonable request.

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
