# Peer review of "Design and Simulate Intracranial Support to Guide Maxillo Surgery: A Study Based on Bioengineering"

_diagnostics, 2023, doi:10.3390/diagnostics13243672_

Round 1
Reviewer 1 Report
Comments and Suggestions for Authors
A techniqal report about techniques to ensure precision in craniofacial surgery.
The work flow is well described but I miss data about the original CT examination on the phantom. The authors test two positions of the sensor (forehead and bite (anterior maxilla)).
The authors report calibrations but it is not described if these calibrations are made by a single person or several persons. No intra- och inter-reability test are reported.
it would be of interest with testing the precision of a sensor placed in the mandible.
Author Response
The entire team of authors thanks the editor-in-chief and the reviewer #1 for their work, appreciation, and advice. Thank you also for your prompt reply. All required changes have been made, they are highlighted in yellow and are best explained below.
The phantom CT scans (axial, coronal, sagittal, and 3D reconstruction) have been inserted into the text as requested by the reviewer – image 1 in rows 62-66 of page 2. The numbering of the other images has been changed.
The calibrations were all performed by the same operator, specified in the text — line 161 of page 5— and for this reason no Inter-rater reliability test was used.
During the conception and design phase, it was proposed to place the device at the level of the jaw. This idea, however, has been abandoned because the jaw is a mobile bone, and this would make both the calibration and the results less reliable.
Reviewer 2 Report
Comments and Suggestions for Authors
This study designed a sensor placement device using an occlusal device to improve the accuracy of maxillary navigation surgery, which may have good application prospects. But there are still some issues that need to be addressed
1. This simulation study was conducted in vitro, and there are many differences from clinical practice. For example, in this study, the sensor on the forehead is adhered to the surface of bone, while in vivo it is adhered to the surface of soft tissue. The movement of soft tissue may affect accuracy, and it is recommended to describe it separately in the discussion.
2. From the statistical results, there is no statistical difference to support the conclusion of this study. It is recommended to re-examine the suitability of the statistical method or modify the current conclusion.
Author Response
The entire team of authors thanks the editor-in-chief and the reviewer #2 for their work, appreciation, and advice. Thank you also for your prompt reply. All required changes have been made, they are highlighted in yellow and are best explained below.
First, let’s say that the proposed study is an in vitro study. Therefore, there are no abnormal sensor movements. The objective is to demonstrate the reproducibility of the device and the study - then the application on the patient will be evaluated. In any case, the sensor is placed on the face of the in vivo patient with a double-sided adhesive sponge. A patch around the head is also used to eliminate any movements of the head and therefore of the sensor.
The conclusion has been amended as suggested. The proposed new methodology, with different sensor positions, has a statistical significance that almost overlaps with the classic method (used to date). What we want to emphasize is that the new proposed device works; moreover, it allows a calibration that is equal to the previous one in some points, while in other points it is better.
Round 2
Reviewer 1 Report
Comments and Suggestions for Authors
I still recommend the authors that two persons perform the calibrations. Then they can calculate both intra- and inter-rater reliability. That would increase the quality of this technical paper.
Author Response
Thank you for your quick response.
Below are the reviews you made:
- Calibrations were carried out by a single researcher to reduce measurement bias.
- The calibrations were all performed by the same operator; hence, the intra-rater reliability test was calculated. Precisely, the consistency of the results by the Pearson correlation coefficient.
All changes are highlighted in yellow in the text.
The order of the first and last author has also been changed (they have only been reversed).

Reviewer 2 Report
Comments and Suggestions for Authors
I think there seems to be a problem with statistical methods, and I suggest consulting statistical experts to assess whether corrections are needed.
Author Response
Thank you for your quick response.
The calibrations were all performed by the same operator; hence, the intra-rater reliability test was calculated. Precisely, the consistency of the results by the Pearson correlation coefficient.
All changes are highlighted in yellow in the text.
The order of the first and last author has also been changed (they have only been reversed).

Round 3
Reviewer 1 Report
Comments and Suggestions for Authors
-
Author Response
All authors would like to thank you for your quick replies and valuable suggestions. Your help was invaluable.
The latest version of the manuscript is attached.

Reviewer 2 Report
Comments and Suggestions for Authors
This version is acceptable
Author Response

(The authors gave the same response as above.)
